# Perioperative Chemotherapy for Liver Metastasis of Colorectal Cancer

**DOI:** 10.3390/cancers12123535

**Published:** 2020-11-26

**Authors:** Gloria Chan, Cheng E. Chee

**Affiliations:** Department of Haematology-Oncology, National University Hospital Singapore, National University Cancer Institute, Singapore 119228, Singapore; gloria_chan@nuhs.edu.sg

**Keywords:** colorectal cancer, liver metastasis, neoadjuvant/adjuvant chemotherapy and conversion chemotherapy

## Abstract

**Simple Summary:**

Survival outcomes for resectable metastatic colorectal cancer have improved over the past decade. This is due in part to improvements made in imaging technology, locoregional treatment, and systemic treatment. The focus of this review is to summarize and analyze the existing information available on systemic therapy in the setting of colorectal cancer with liver metastases. It is written with structured viewpoints including patient selection, sequencing of treatment, choice of systemic therapy for upfront resectable disease and conversion chemotherapy for unresectable colorectal cancer. Post-treatment surveillance and future research directions are also discussed. This review will help treating physicians make decisions for the treatment of liver metastases of colorectal cancer.

**Abstract:**

The liver is the dominant site of metastasis for patients with colorectal cancer. For those with isolated liver metastases, surgical resection with systemic therapy has led to long-term remission in as high as 80% of patients in well-selected cohorts. This review will focus on how systemic therapy should be integrated with resection of liver metastases; in particular, the use of clinical risk scores based on clinicopathological features that help with patient selection, various approaches to the treatment of micro-metastatic disease (peri-operative versus post-operative chemotherapy), as well as conversion chemotherapy for those with initially upfront unresectable disease will be discussed.

## 1. Introduction

According to the World Health Organization (WHO) Global Cancer Observatory (GLOBOCAN) 2018 data, colorectal cancer (CRC) is the third most common cancer diagnosed (10.2%) and the second-leading cause of mortality amongst all cancers (9.2%) [1]. Over 1.8 million people worldwide were diagnosed with CRC in 2018, 15 to 25% of them would have synchronous liver metastases, and another 20% would develop liver metastases within three years, from the time of the first diagnosis [2,3,4]. The five-year overall survival (OS) in all stage IV CRC is 14.3% [5]. This is in stark contrast to what is observed for patients with resectable liver metastases, whose OS has improved remarkably over the last two decades. In fact, five-year OS from databases around the turn of the millennium was reported to be between 30 and 40% [6,7,8], compared to figures higher than 80% in trials published in 2020 [9].

## 2. Patient Selection

As there is a clear survival impact, surgical resection of R0 resectable liver metastases is the treatment of choice when possible. The criteria to select patients for potentially curative resection is constantly being redefined, and this varies between institutions but is largely dependent on the experience of the multi-disciplinary medical team. Considerations that need to be accounted for include patient factors, disease factors, and anatomic factors.

There are several well-established clinical risk scores that are used to stratify patients based on their likelihood of recurrence (Table 1) [7,10,11,12]. To spare patients from a futile operation, those with high-risk features should be considered for initial chemotherapy to assess tumor biology before surgery.

Even though these scores have been widely used to stratify patients based on their likelihood of recurrence, they were developed based on patients who underwent surgery before 2000, when prognostic information of tumor mutations were not known. Multiple groups have shown that somatic mutations in RAS are associated with inferior progression-free and overall survival in colorectal cancer [13,14]. Retrospective data based on 1460 patients who underwent colorectal liver metastasis resection and multigene testing revealed that RAS mutation status alone is not sufficient for predicting prognosis after resection [15]. These patients should not be uniformly excluded from surgery as long as a R0 resection can be achieved, as it remains the only option for cure [16,17,18,19]. Tumors with BRAF V600E mutations are associated with aggressive tumor biology and are less likely to be candidates for metastatectomy due to widespread disease at the time of presentation. Median OS even after resection of liver metastases for this group of patients is between 22 to 28 months [20,21]. Therefore, resection of liver metastases in this group of patients should only be considered if there has been a durable response to systemic therapy and the liver was the only site of disease.

More recently validated risk scores have included factors like KRAS mutation status and presence of extrahepatic disease, as in the Genetic and Morphological Evaluation (GAME) score [22], and KRAS/NRAS/BRAF mutation status and presence of bi-lobar liver metastasis in the comprehensive evaluation of relapse risk (CERR) score, both of which have outperformed the initial Fong Criteria [23]. Anatomic criteria when selecting candidates for resection of colorectal liver metastases have broaden so much over the years that the modern consensus on what defines resectable liver metastases is tumors that can be resected completely while leaving adequate liver remnant [24]. All in all, no set of guidelines can replace case-by-case evaluation by a multi-disciplinary team, taking into consideration the complex web of interactions between “resectability”, tumor biology, and patient factors; this remains the standard of care.

## 3. Timing of Systemic and Surgical Therapy

The optimal timing and sequencing of chemotherapy, chemoradiotherapy (for patients with a rectal primary), and surgery should be individualized for each patient. This decision is undertaken by a multi-disciplinary tumor board and should account for the following: synchronous versus metachronous liver metastases, presence of a symptomatic primary tumor, resectability of liver metastasis, and response to systemic therapy. There is no evidence that adding a biological agent to a cytotoxic doublet improves the outcome in the presence of resectable metastases compared with a cytotoxic doublet alone in combination with resection of the metastases. In general, patients with a symptomatic primary tumor (e.g., bleeding, obstruction, perforation) should have treatment of the primary tumor prioritized. If liver metastases are upfront resectable, concurrent resection of liver metastases can be undertaken if the peri-operative risks are acceptable. If not, pre-operative chemotherapy may be given, followed by reassessment with close-interval imaging. It is important to proceed to liver metastatectomy once resection is deemed feasible and not to continue with prolonged chemotherapy to avoid veno-occlusive disease or chemotherapy-associated steatohepatitis. Those with upfront resectable metachronous liver metastases (i.e., metastatic recurrence in the liver following resection of the primary tumor) and low clinical risk of recurrence can be considered for upfront resection. Following resection, the benefit of adjuvant chemotherapy or surveillance alone will need to be discussed with the patient. Figure 1 summarizes the approach to colorectal liver metastases.

### 3.1. Peri-Operative Chemotherapy for Upfront Resectable Disease

In those with upfront resectable disease, studies have looked at delivering systemic chemotherapy either for six months post-operatively or in a sandwich fashion, i.e., three months before and three months after resection of hepatic metastases. In those with resectable disease, the goal of delivering chemotherapy pre-operatively is to “test tumor biology”, that is, to allow an observational period to determine whether new lesions will appear soon after systemic treatment, rendering surgery futile. Conversely, we know that the pathologic response to pre-operative chemotherapy is predictive of prognosis after resection of colorectal liver metastases; the five-year OS in those attaining pathological complete response is 76%, compared to 45% in those who do not [25].

To date (Table 2), there are two Phase III clinical trials that randomized patients in the sandwich regimen—the EORTC 40983 and the New EPOC study. In the first study, 364 patients with CRC and resectable liver metastases (up to four) were randomized to surgery alone or six cycles of FOLFOX4 chemotherapy (Fluorouracil, Leucovorin, Oxaliplatin), administered before and after surgery [26]. Three-year progression-free survival (PFS) did not increase significantly with the addition of chemotherapy. However, the improvement of 8% in all eligible patients and 9% in patients who had resection was significant, even though not pre-specified. Pre-operative chemotherapy achieved a response rate of 43%, 7% of the patients had progressive disease, and eight subjects (5%) were inoperable due to disease progression. Tumor progression on chemotherapy predicts poor outcomes after resection [27], and pre-operative chemotherapy helps to select patients who would otherwise go through unnecessary surgery. Opponents of pre-operative chemotherapy would argue that it delays otherwise curative surgery and increases the risk of post-operative complications [28]. Although surgery is deemed as curative in this setting, it has been shown that in those patients who achieve complete radiological response (3%) with chemotherapy, viable tumors are still observed in the resected liver specimen in more than 80% of these cases, which could have precluded them from resection in the first place [29]. Furthermore, there was no observed OS difference for patients treated with peri-operative FOLFOX4 during long-term follow-up. The five-year OS was 51.2% for patients in the chemotherapy arm versus 47.8% for the surgery-only group [30].

Although the addition of anti-epidermal growth factor receptor (EGFR) antibodies such as cetuximab and panitumumab have shown to produce a survival benefit for patients with RAS/RAF wild-type tumors in the palliative setting [31,32], this is not the case for patients with operable liver metastases. The New EPOC trial randomized patients with KRAS exon 2 wild-type tumors with resectable or borderline resectable liver metastases to receive chemotherapy (oxaliplatin plus fluorouracil, oxaliplatin plus capecitabine, or irinotecan plus fluorouracil) with or without cetuximab before and after resection of liver metastases. Despite a higher number of patients showing response to treatment with cetuximab, the trial was terminated due to the detrimental effect on PFS that was observed during interim analysis, which eventually led to a shortening of OS of 26 months in the group receiving cetuximab [33,34]. The effects of anti-EGFR therapy in the setting of resectable metastases, such as tumor sidedness, RAF mutation, HER2 amplification, and microsatellite instability status, which were untested at the time of trial enrolment as the knowledge of resistance to anti-EGFR antibodies, post-date the design of this trial [35]. However, mutation status alone would not explain the result of this trial, as post-hoc analysis showed no significant difference in the distribution of extended RAS and RAF mutations in both groups. Interestingly, cetuximab was more detrimental in subgroups that were associated with good prognostic features such as well- or moderately differentiated primary tumors, fewer number of liver metastases, lack of N2 disease and metachronous disease [33]. Post-recurrence survival was much worse in the group that received cetuximab, possibly suggesting an aggressive disease phenotype development at recurrence or imbalances in post-recurrence treatment approaches [36]. Only 10% of patients who received peri-operative cetuximab received it again in the recurrent setting, compared to 30% in the chemotherapy-only arm. The findings of the New EPOC trial echo the results of N0147 and PETACC-8 in the adjuvant setting that saw no benefit with the addition of cetuximab to oxaliplatin-based chemotherapy in patients with stage III disease [37,38]. Overall, the consistent trend of lack of efficacy (and perhaps even the presence of a detrimental effect) of anti-EGFR therapy in patients with micro-metastatic disease makes this treatment unsuitable for use in the curative setting.

### 3.2. Adjuvant Chemotherapy

The use of oxaliplatin-based adjuvant chemotherapy for the treatment of micro-metastatic disease has long been accepted as the standard of care for patients with stage 3 colon cancer due to the consistent, significant disease-free and overall survival benefits [39]. However, this benefit cannot be directly extrapolated to post-operative chemotherapy after resection of liver metastases. Several trials designed to answer this question have been fraught with issues of poor recruitment, early termination, and usage of non-modern chemotherapy by today’s standards [40,41,42]. Table 3 summarizes the randomized trials for adjuvant fluorouracil-based chemotherapy after resection of liver metastases. In three studies, fluorouracil-based chemotherapy was compared to observation [40,41,43], showing a consistent improvement in PFS of 6–9% with adjuvant chemotherapy. However, this improvement did not consistently translate to a significant improvement in OS at longer term follow-up. The preliminary data from JCOG0603, which was presented at the American Society of Medical Oncology (ASCO) meeting in 2020, represent the most recent results we have on chemotherapy after resection of liver metastases. The trial compared six months of post-operative mFOLFOX6 with observation alone and found a significant improvement in three-year disease-free survival (DFS) of 10.6% with chemotherapy. However, this trial was terminated early because of the futility in OS; the five-year OS was 83.0% in the control arm compared to 69.5% for patients receiving chemotherapy. Only 44% of those receiving adjuvant chemotherapy completed the six-month trial. The reasons suggested for the detrimental effect of mFOLFOX6 on OS was the restricted use of oxaliplatin for recurrent disease and the emergence of more aggressive chemotherapy-refractory tumors at relapse in those patients who received post-operative chemotherapy [9]. The results of this study also suggest that a five-year OS of 83% is possible after resection of small (<5 cm) and a limited number (≤3) of liver metastases followed by surveillance alone.

### 3.3. Conversion Chemotherapy for Unresectable Disease

The purported clinical benefits are seen only in R0 resection (i.e., no gross or microscopic tumor remains in the primary tumor bed); there is no role for partial palliative resection of metastases. As such, the consideration of upfront chemotherapy in the presence of initially unresectable liver metastases with the hope of downstaging to resectable disease seems to be a reasonable approach, but in reality it only occurs in 12% of cases [45]. However, for those who achieve a good response and successfully undergo curative resection treatment, the five-year DFS rate is 22% [45]. In choosing this approach, consideration must be given to the timing of chemotherapy, as prolonged administration of chemotherapy may lead to higher risk of liver toxicity and post-operative morbidity [46].

Multiple regimens for conversion chemotherapy in the presence of unresectable CRC with liver metastases have been studied, but the optimal chemotherapy regimen has not yet been established (Table 4). Broadly, the treatment should involve doublet or triplet chemotherapy with or without addition of targeted therapy. Factors considered include need for response, sidedness of primary tumor, mutation status, and previous chemotherapy-related toxicities.

Oxaliplatin-based (FOLFOX/XELOX) and irinotecan-based doublets (FOLFIRI/XELIRI) result in similar response rates, between 34% and 59% [47,48,49,50,51,52,53,54]. A careful selection of patients for conversion chemotherapy in Phase II clinical trials showed that the approach with doublet chemotherapy results in an R0 resection rate of 24% to 40% [48,52,53]. Further intensification to a triplet regimen with FOLFOXIRI resulted in higher responses (60% to 70%) and in a 15–26% rate of R0 resection [49,55]. In deciding the duration of chemotherapy pre-operatively, one should bear in mind that the purpose of conversion chemotherapy is not to treat until maximal response but rather to provide a limited course of chemotherapy until response has occurred to enable liver resection [56]. The incremental risk of post-operative morbidity is directly related to the number of chemotherapy cycles administered before surgery [28]. Cytotoxic chemotherapy used for metastatic CRC is associated with sinusoidal dilation in the case of oxaliplatin and steatohepatitis in the case of irinotecan, which increased the odds of post-operative mortality by ten-fold [46].

Bevacizumab is a humanized monoclonal antibody against vascular endothelial growth factor (VEGF) which has shown survival benefit in the palliative setting [57,58,59]. The addition of bevacizumab to either doublet or triplet chemotherapy has been investigated in several trials [50,60,61,62,63,64]. Results from a pooled analysis of 11 trials including 889 patients showed that a FOLFOXIRI–bevacizumab combination resulted in response rates of 69% and R0 resection in 28% of those with initially had unresectable metastases [65]. Yet, the addition of bevacizumab to chemotherapy does not appear to significantly increase the response rates or rate of R0 resection in the NO16966 study [50]. Bevacizumab belongs to a class of anti-angiogenic agents with a peculiar toxicity profile including hypertension, proteinuria, hemorrhage, thromboembolism, gastrointestinal perforation, and impaired wound healing [66]. The half-life of bevacizumab is around 20 days, and the current data suggest that an interval of 5 to 8 weeks from the last dose of bevacizumab to surgery is likely to be safe [67,68]. However, given that bevacizumab has not shown to consistently improve response rates [50], does not provide benefit in the adjuvant setting [69,70], and has well-established side effects, it may not need to be routinely administered for the goal of conversion.

Addition of anti-EGFR agents (cetuximab or panitumumab) to doublet chemotherapy in the setting of unresectable liver metastases results in response rates of 41–70%, translating to R0 resection rates of 27–38% [31,71,72,73,74]. In the CELIM trial, cetuximab added to either FOLFIRI or FOLFOX significantly increased the rate of R0 resection (25% vs. 7%) compared to chemotherapy alone [75]. The combination of EGFR inhibitors to triplet chemotherapy has consistently shown better responses, compared to FOLFOXIRI in patients with wild-type RAS tumors [63,76]. Patients with metastatic wild-type RAS CRC were randomized to receive panitumumab plus mFOLFOXIRI vs. mFOLFOXIRI only in the VOLFI trial [76]. The arm receiving quadruplet therapy achieved an overall response rate (ORR) of 86%, while those administered triplet therapy had an ORR of 54%. Among those receiving panitumumab, ORR was significantly higher in patients with left-sided tumors compared to those with right-sided tumors (90% vs. 60%). For those patients where an eventual resection was planned, more patients receiving panitumumab achieved resection (60% vs. 36%). Apart from the RAS/RAF molecular status, this trial and several others indicate that primary tumor sidedness is predictive for response to EGFR inhibitors [77,78]. In deciding between the use of anti-VEGF or an anti-EGFR, the sidedness of the primary tumor needs to be considered. In a combined analysis of six clinical trials (CRYSTAL [31], FIRE-3 [79], CALGB 80405 [80], PRIME [32], PEAK [81], and 20050181 [82]), significant improvements in PFS and OS with the addition of cetuximab or panitumumab to chemotherapy were only seen in patients with left-sided (distal to the splenic flexure), wild-type RAS primary colorectal tumors [78]. For the response rates, there was a trend towards a greater benefit from anti-EGFR therapy in those with left-sided tumors compared to those with right-sided tumors [78]. The National Comprehensive Cancer Network (NCCN) and the European Society of Medical Oncology (ESMO) recommend that the incorporation of EGFR inhibitors in the first-line setting should be indicated only for those patients with wild-type RAS/RAF and left-sided tumors [83,84]. Contrastingly, for patients with more aggressive right-sided primary tumors, addition of bevacizumab to chemotherapy may be more beneficial.

## 4. Surveillance

Post-treatment surveillance is indicated for patients who would be considered for a second potentially curative surgical procedure. Approximately 60% of cancers recur a year after the complete resection of colorectal liver metastases, and about one-quarter of patients who undergo surveillance after initial treatment of liver metastases can undergo curative-intent treatment again [85]. The five-year DFS and OS for those who undergo repeat hepatectomy have been reported as 22–26% and 41–73%, respectively [86,87]. NCCN recommends CEA testing and contrast-enhanced CT scans of the thorax, abdomen, and pelvis every 3–6 months in the first two years after adjuvant chemotherapy and then every 6–12 months for a total five years [83].

## 5. Future Directions

This year, the Federal Drug Agency (FDA) approved anti-PD1 pembrolizumab for first-line treatment of patients with metastatic CRC with microsatellite instability-high (MSI-H) or mismatch repair-deficient (dMMR) CRC based on the findings of the KEYNOTE-177 study [88]. Compared to chemotherapy (FOLFOX or FOLFIRI with bevacizumab or cetuximab), first-line pembrolizumab resulted in doubling of PFS (16 vs. 8 months) and improved ORR (44% vs. 33%) [89]. In the updated analysis of CheckMate 142, the combination of nivolumab and low-dose ipilimumab achieved an ORR of 64%, a complete response rate of 9%, and a disease control rate of 84% [90]. Although the dMMR status is uncommon (<5% of metastatic CRC patients), the high and durable response seen with checkpoint inhibitors in this group of patients raises the question of liver metastatectomy in cases which are converted to resectable disease. Interestingly, a small retrospective study observed pathologic complete response in the majority of resected specimens which were treated with checkpoint inhibitors, despite presence of residual tumors on pre-operative imaging [91]. This suggests that residual radiographic tumors may not require resection following response to anti-PD1-based therapy. Hence, the role of liver metastatectomy in dMMR patients remains uncertain and needs to be prospectively validated.

The lack of OS benefit with modern post-operative chemotherapy after liver resection indicates that patient selection needs improvement. One emerging area is the potential use of circulating tumor DNA (ctDNA), as it defines minimal residual disease, reflecting the existence of micro-metastases after surgical resection [92]. Following hepatectomy for liver metastases in CRC, detection of ctDNA was associated with a high risk of recurrence [93]. In future post-operative studies, ctDNA could be used to select patients with a high risk of recurrence and be considered for the escalation of chemotherapy regimens. ctDNA has also been suggested to help select patients for resection of colorectal liver metastases, in that the absence of ctDNA after four weeks of systemic chemotherapy correlated with an 85% R0/R1 resection rate of liver metastases [94].

## 6. Conclusions

In conclusion, the management of CRC liver metastasis requires risk stratification and a multi-disciplinary input. Those patients with upfront resectable disease and low clinical risk should undergo resection followed by a risk–benefit discussion regarding adjuvant chemotherapy post-operatively or surveillance. In patients who fulfill the criteria of the EORTC 40983 study, peri-operative chemotherapy may be considered. The peri-operative chemotherapy approach may also be considered for resectable but high-risk cases to test tumor biology in order to avoid futile surgery. For unresectable liver metastases, a good response to systemic therapy may provide an opportunity for liver metastatectomy. It is important to note that the goal of conversion chemotherapy is surgical resection rather than maximal response; hence, regular imaging at close intervals is required to determine the optimal time for resection in order to reduce the risk of chemotherapy-induced liver injury. The choice of treatment regimen for conversion chemotherapy depends on tumor burden, RAS/RAF mutation status, primary tumor sidedness, exposure to previous adjuvant chemotherapy, and pre-existing toxicities. Regardless of the regimen chosen prior to resection, a total of six months of peri-operative systemic therapy is recommended. Chemotherapy based on 5FU with or without oxaliplatin is recommended in the post-operative setting, as there is no proven benefit for irinotecan, bevacizumab, or EGFR inhibitors when there is no evaluable disease. More reliable biomarkers are required in this setting to better select patients for treatment in order to optimize the current standards of care.

## Figures and Tables

**Figure 1 cancers-12-03535-f001:**
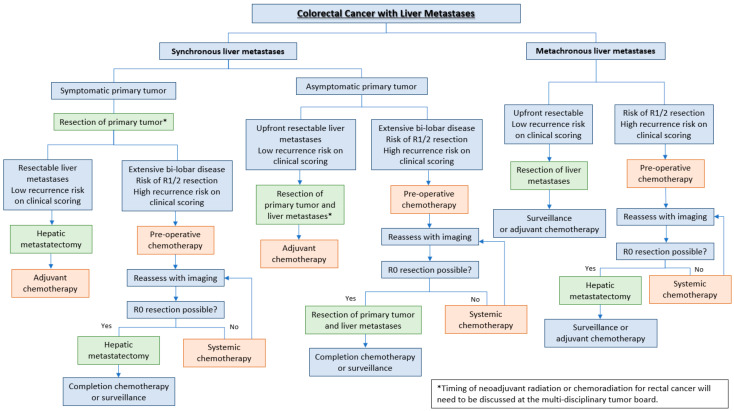
Decision tree for timing and sequence of systemic therapy and surgery for patients with metastatic colorectal cancer to the liver.

**Table 1 cancers-12-03535-t001:** Clinical risk criteria for assessing the risk of recurrence.

Criteria *		Fong [7]	Nordlinger [10]	Nagashima [11]	Konopke [12]
Disease-free interval		<12 months	<24 months	-	Synchronous liver metastases
No. of metastases		>1	>3	>1	>3
Largest liver metastases		>5 cm	>5 cm	>5 cm	-
Lymph node staging		≥N1	≥N1	≥N1	-
Tumor staging		-	pT4	pT4	-
Pre-op CEA (ng/mL)		>200	-	-	≥200
Age (years)		-	>60	-	-
Risk groups (scores)	Low	0–2	0–2	0–1	0
	Intermediate	-	3–4	2–3	1
	High	3–5	5–6	≥4	≥2

* 1 point assigned for each risk factor, CEA = carcinoembryonic antigen.

**Table 2 cancers-12-03535-t002:** Peri-operative chemotherapy for resectable disease. PFS, progression-free survival, OS, overall survival, FOLFOX4, Fluorouracil, Leucovorin, Oxaliplatin.

Author, Year	N	Arms	Response Rates	PFS Outcomes (Months)	OS Outcomes (Months)
Nordlinger, 2008, 2013 [26,30]	364	FOLFOX4	43%	Median (m) PFS: 18.7 v. 11.7 3-year PFS^: 35.4 (28.1–42.7) v. 28.1 (21.3–35.3)	mOS: 61.4 (51.0–83.4) v. 54.3 (41.9–79.4) 5-yr OS (%): 51.2 (43.6–58.3) v. 47.8 (40.3–55.0)
		Observation	-		
Primrose, 2014 [33] Bridgewater, 2020 [34]	257	Chemotherapy ^1^ + Cetuximab	70%	mPFS: 15.5 (13.0–19.0) v. 22.2 (18.3–26.8)	mOS: 55.4 (43.5–71.5) v. 81.0 (59.6-NR)
		Chemotherapy ^1^	62%		

^1^ Chemotherapy: oxaliplatin plus fluorouracil, oxaliplatin plus capecitabine, or irinotecan plus fluorouracil; ^ Did not reach statistical significance.

**Table 3 cancers-12-03535-t003:** Adjuvant chemotherapy for resected disease. DFS, disease-free survival.

Author, Year	N	Arms	DFS	OS
Portier, 2006 [40]	173 (aim = 200)	5FU/LV	mDFS: 24.4 (SE = 3.6) v. 17.6 (SE = 2.7) months [∆ = 6.8 months] 5-year DFS: 33.5% v. 26.7% [∆ = 6.8%]	mOS ^: 62.1 (SE 10.7) v. 46.4 (SE 4.6) months [∆ = 15.7 months] 5-year OS ^: 51.1% v. 41.1% [∆ = 10.0%]
		Observation		
Mitry, 2008 [41]	278 pooled analysis of 2 trials FFCD ENG	5FU/LV	mDFS ^: 27.9 v.18.8 months [∆ = 9.1 months] 5-year DFS ^: 36.7 (24.5–41.1) v. 27.7 (20.0–35.9) [∆ = 9.0%]	mOS ^: 62.2 vs. 47.3 months [∆ = 14.9 months] 5-year OS ^: 52.8% (43.7–61.3) v. 39.6 (30.7 to 48.3) [∆ = 13.2%]
		Observation		
Ychou, 2009 [44]	306 (aim = 420)	FOLFIRI	mDFS: 24.7 v. 21.6 months [∆ = 3.1] 2-year DFS ^: 46% v. 51%	3-year OS ^: 72% v. 73%
		5FU/LV		
Kobayashi, 2014 [43]	180	UFT/LV	3-year DFS: 38.6% v. 32.3% [∆ = 6.3%]	3-year OS ^: 82.8% v, 81.6%
		Observation		
Kanemitsu, 2020 [9]	300	mFOLFOX6	3-year DFS: 52.1% (43.2–60.2) v. 41.5% (33.2–49.6) [∆ = 10.6%]	5-year OS ^: 69.5% (59.6–77.5) v. 83.0% (74.5–88.9) mOS after recurrence: 38.4 v. 87.6 months
		Observation		

5FU: Fluorouracil; FOLFIRI: fluorouracil, irinotecan; UFT: uracil/tegafur; LV: leucovorin; mFOLFOX6: Fluorouracil, Oxaliplatin; ^ Did not reach statistical significance.

**Table 4 cancers-12-03535-t004:** Regimens for conversion chemotherapy for patients with colorectal cancer with liver metastases.

Class	Regimens	Trials	ORR (%)	Resection Rate (%)
Chemotherapy-only regimens	FOLFOX	Delaunoit, 2005 [47]	54	3
		Alberts, 2005 [48]	59	40
		Falcone, 2007 [49]	34	6 (R0)
		Saltz, 2008 (NO16966) [50]	47	6
		Bokemeyer, 2008 (OPUS) [73]	37	4 (R0)
		Douillard, 2010 (PRIME)	48	7 (R0)
	XELOX	Saltz,2008 (NO16966) [50]	47	6
	FOLFIRI	Barone, 2007 [54]	47	32
		Skof, 2009 [52]	48	24 (R0)
		Pozzo, 2004 [53]	47	32 (R0)
		Van Custem, 2009 (CRYSTAL) [31]	38	1.7 (R0)
	FOLFOX/FOLFIRI	Ye, 2013 [75]	29	7.4 (R0)
	XELIRI	Skof, 2009 [52]	49	24 (R0)
	FOLFOXIRI	Falcone, 2007 [49]	60	15 (R0)
		Geissler, 2017 (VOLFI) [76]	54	36
		Folprecht, 2020 (AIO-CELIM2) [63]	72	-
		Ychou, 2008 [55]	70	26 (R0)
Cytotoxics + Anti-EGFR	FOLFOX + Cetuximab	Folprecht, 2010 [71]	70	38 (R0)
		Ji, 2013 [72]	73	27 (R0)
		Bokemeyer, 2008 (OPUS) [73]	61	10 (R0)
	FOLFIRI + Cetuximab	Folprecht, 2010 [71]	41	30 (R0)
		Van Custem, 2009 (CRYSTAL) [31]	47	4.8 (R0)
		Folprecht, 2020 (AIO-CELIM2) [63]	81	-
	FOLFOX/FOLFIRI + Cetuximab	Ye, 2013 [75]	57	26 (R0)
	FOLFOXIRI + Cetuximab	Folprecht, 2020 (AIO-CELIM2)	86	-
	FOLFOX + Panitumumab	Douillard, 2010 (PRIME) [74]	55	8.3 (R0)
	FOLFOXIRI + Panitumumab	Geissler, 2017 [76]	86	60
Cytotoxics + Bevacizumab	FOLFOX + Bevacizumab	Saltz, 2008 (NO16966) [50]	49	8.4
		Gruenberger, 2015 (OLIVIA) [60]	62	23 (R0)
	XELOX + Bevacizumab	Saltz, 2008 (NO16966) [50]	49	8.4
		Wong, 2011 [64]	78	47.8
		Gruenberger, 2008 [67]	73	92.9 (R0)
	FOLFIRI + Bevacizumab	Loupakis, 2014 (TRIBE) [61]	53	12 (R0)
	FOLFOXIRI + Bevacizumab	Masi, 2010 [62]	77	26 (R0)
		Loupakis, 2014 (TRIBE) [61]	65	15 (R0)
		Gruenberger, 2015 (OLIVIA) [60]	81	49 (R0)
		Folprecht, 2020 (AIO-CELIM2) [63]	70	-

FOLFOX: Fluorouracil, Oxaliplatin; XELOX: Capecitabine, Oxaliplatin; FOLFIRI: Fluorouracil, Irinotecan; XELIRI: Capecitabine, Irinotecan; FOLFOXIRI: Fluorouracil, Oxaliplatin, Irinotecan.

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
