# Peer review of "Perioperative Chemotherapy for Liver Metastasis of Colorectal Cancer"

_cancers, 2020, doi:10.3390/cancers12123535_

Round 1
Reviewer 1 Report
A brief summary
This review highlights the clinical significance of perioperative chemotherapy integrated with surgical resection for liver metastasis of colorectal cancer. This review also emphasizes the importance of patient-tailored treatment strategy for synchronous or metachronous liver metastasis of colorectal cancer based on individual recurrence risk stratifications including anatomy, patient characteristics, and tumor biology.
Broad comments
This review is well written with structured viewpoints including patient selection, perioperative chemotherapy for upfront resectable liver metastases, adjuvant chemotherapy, conversion chemotherapy for unresectable disease, surveillance, and future directions with ongoing studies and updated information on molecular markers.
This review may contribute to improved knowledge of recent trends and help make decisions in the treatment of liver metastasis of colorectal cancer, for both medical oncologists and surgeons.
Specific comments
None
Author Response
Thank you for your comments!
I have included your comments in the simple summary provided in the revised version.
"Survival outcomes for resectable metastatic colorectal cancer have improved over the past decade. This is due in part to improvements made in imaging technology, locoregional treatment and systemic treatment. The focus of this review is to summarize and analyze the existing information available on systemic therapy in the setting of colorectal cancer with liver metastases. It is written with structured viewpoints including patient selection, sequencing of treatment, choice of systemic therapy in both upfront resectable disease as well as conversion chemotherapy for unresectable colorectal cancer. Post-treatment surveillance and future research directions are also discussed. It will help treating physicians make decisions in the treatment of liver metastasis of colorectal cancer."
Reviewer 2 Report
This manuscript entitled “Perioperative Chemotherapy for Liver Metastasis of Colorectal Cancer” by Gloria HJ Chan Cheng E. Chee reviewed the current update about perioperative chemotherapy for patients with mCRC and liver metastases. Overall, the review is well-written and summarize the most update clinical trials and data. I have some suggestions for this review. Some comments: 1. Lines 10-15? 2. Line 27 GLOBOCAN 2018 stand for? 3. Line 34 “compared to figures higher than 80% in trials published 34 in 2020 [9].” The reference cited seems not compatible with what authors state. 4. Figure 1. For patients with synchronous liver metastases, if R0 resection is not possible, systemic chemotherapy is suggested. But reassessment should be suggested after chemotherapy as suggestion from metachronous liver metastases. 5. Primary tumor resection is not needed for asymptomatic primary tumor? 6. Line 124. The authors should comment why this trial failed (New EPOC trial). 7. Line 150. “this improvement never translated to a significant improvement in OS at longer term follow-up.” References 41, 42 are not compatible with this statement?
Author Response
Thank you for your comments.
Lines 10-15. The plain text summary has been included: Survival outcomes for resectable metastatic colorectal cancer have improved over the past decade. This is due in part to improvements made in imaging technology, locoregional treatment and systemic treatment. The focus of this review is to summarize and analyze the existing information available on systemic therapy in the setting of colorectal cancer with liver metastases. It is written with structured viewpoints including patient selection, sequencing of treatment, choice of systemic therapy in both upfront resectable disease as well as conversion chemotherapy for unresectable colorectal cancer. Post-treatment surveillance and future research directions are also discussed. It will help treating physicians make decisions in the treatment of liver metastasis of colorectal cancer.
Line 26. Abbreviation GLOBCAN has been edited. It now reads, “According to the World Health Organization (WHO) Global Cancer Observatory (GLOBOCAN) 2018 data, colorectal cancer (CRC) is the third most common cancer diagnosed (10.2%) and the second-leading cause of mortality amongst all cancers (9.2%) [1].”
Line 34. 5-year OS from databases around the turn of the millennium was reported to be between 30 to 40% [6–8], compared to figures higher than 80% in trials published in 2020 [9]. We changed the reference to the JCOG0603 study presented at ASCO 2020, reporting median overall survival of 83% (95% CI 74.5-88.9) in the arm who had surgery alone, without post-operative chemotherapy. The reference provided previously was wrong. Thanks for highlighting!
Figure 1. Thanks for your input! We have included reassessment after chemotherapy as well. You’re right. Primary tumour should be resected in the curative setting. We have included this as well.
Lines 124-133. The detrimental effects of addition of cetuximab in the New EPOC trial have been included. Apart from RAS/RAF mutations that were initially untested, it has been suggested that in those exposed to cetuximab, a more aggressive disease phenotype develops at recurrence. It may also be due to imbalance of post-recurrence treatment. Only 10% of patients in the cetuximab arm received anti-EGFR at recurrence, compared to 30% in those who did not receive cetuximab prior.
Lines 144-145. Both trials referenced had an improvement in DFS, numerically improved OS, but did not reach statistical significance. The values are included in Table 3.
Thank you very much for your comments once again!
Reviewer 3 Report
excelent review, well aritten and refrenced
Author Response
Thank you for your comments! Minor edits have been made to the revised paper.